# Intensive Cognitive Behavioral Therapy for Adolescents with Anorexia Nervosa Outcomes before, during and after the COVID-19 Crisis

**DOI:** 10.3390/nu16101411

**Published:** 2024-05-08

**Authors:** Riccardo Dalle Grave, Mirko Chimini, Gianmatteo Cattaneo, Anna Dalle Grave, Loretta Ferretti, Sofia Parolini, Simona Calugi

**Affiliations:** Department of Eating and Weight Disorders, Villa Garda Hospital, 37016 Verona, Italy; mirko.chimini1@gmail.com (M.C.); gianmatteocattaneo@gmail.com (G.C.); annadallegrave@gmail.com (A.D.G.); loretta.ferretti16@gmail.com (L.F.); sofia.parolini14@gmail.com (S.P.); si.calugi@gmail.com (S.C.)

**Keywords:** anorexia nervosa, adolescent, eating disorder, cognitive behavioral therapy, COVID-19 pandemic

## Abstract

Studies comparing treatment outcomes in patients with eating disorders before and during the coronavirus (COVID-19) pandemic have yielded conflicting results. Furthermore, no study has yet evaluated treatment outcomes in adolescent patients with anorexia nervosa before, during and after the crisis. Hence, this study investigated the outcomes of an intensive Cognitive Behavioral Therapy-Enhanced (CBT-E) program on adolescents with anorexia nervosa consecutively treated before (n = 64), during (n = 37) and after (n = 31) the period of emergency spanning 8 March 2020 to 31 March 2022. Results show consistent and similar improvements in eating disorder psychopathology, general psychopathology and body mass index-for-age percentiles across all three periods, with approximately 60% of patients maintaining a full response at the 20-week follow-up, suggesting that treatment efficacy remained robust. Overall, the study underscores the effectiveness of intensive CBT-E as a viable treatment option for adolescents with anorexia nervosa, even during and after unprecedented challenges such as those posed by the COVID-19 pandemic.

## 1. Introduction

The global outbreak of the coronavirus disease (COVID-19) led governments worldwide to implement various measures aimed at mitigating its spread. The adoption of these social and physical distancing measures, driven by concerns over the potential severe consequences of COVID-19 infection, subjected individuals to numerous stressors. Among those were heightened social isolation, disruption to daily routine, limitations on movement, strained relationships with family and friends and increased exposure to weight-related content on social media. These stressors have been linked to a rise in the incidence of eating disorders [1] and an increase in the number of individuals seeking out- and inpatient treatment for eating disorders [2,3,4,5], particularly among adolescents.

Moreover, as a result of the restrictions, outpatient and inpatient services for eating disorders had to find innovative ways of coping with the increased demand, often transitioning to remote delivery [6]. In addition, even when in-person services have been feasible, their quality has been adversely affected by concerns about infection, the necessity for personal protective equipment, mandated social distancing measures and stringent disinfection protocols [7]. Additionally, inpatient eating disorder units previously organized as “open” units have had to be restructured into “closed” units [8].

While such alterations to treatment implementation have been linked to observed deteriorations in eating disorder and overall psychopathology scores among patients with eating disorders, studies comparing outcomes in patients treated before and during the COVID-19 pandemic have yielded conflicting results. Bracké et al. [9], for example, found that patients with anorexia nervosa recruited from mental health institutions and hospitals and treated during the pandemic showed a lower increase in body mass index (BMI) but a similar change in eating disorder symptoms over time than pre-pandemic patients. Similar findings were yielded by a study comparing face-to-face pre-pandemic and online during-pandemic treatment in patients of the Step Up or Day Care Eating Disorder Day Services, specifically that online treatment led to significant improvements in eating disorder psychopathology but not BMI [10].

In contrast, other studies have found that patients with eating disorders showed the expected improvements in eating disorder psychopathology and weight gain [11,12]. For instance, one study investigated the outcomes of intensive enhanced cognitive behavioral therapy for eating disorders (CBT-E) in a cohort of 57 older adolescents and adult patients with anorexia nervosa who experienced the COVID-19 pandemic during treatment as compared with a control group composed of gender-, age- and BMI-matched patients with anorexia nervosa receiving the same treatment before the pandemic. Both groups exhibited significant improvements in BMI, eating disorder and general psychopathology, and clinical impairment scores from baseline to the 20-week follow-up. However, the improvement was more pronounced in the pre-pandemic control group than those treated during the COVID-19 pandemic [8].

In light of these findings, more rigorous investigations into treatment outcomes in patients with anorexia nervosa treated during the COVID-19 pandemic are warranted. Moreover, to date, no study has assessed whether the deterioration in outcomes of intensive treatment for eating disorders reported by some has persisted after the COVID-19 emergency. Hence, the purpose of this study was to compare the outcomes of intensive CBT-E in consecutive adolescent patients with anorexia nervosa who completed the treatment before, during and after the COVID-19 pandemic. The ultimate aim was to determine whether it is necessary to design specific strategies and procedures to better help patients during this post-pandemic period.

## 2. Materials and Methods

### 2.1. Study Design and Participants

This cohort study was conducted on a sample of adolescent patients with anorexia nervosa consecutively admitted to the Villa Garda Hospital Department of Eating and Weight Disorders, Italy, before, during and after the COVID-19 emergency (Figure 1).

All participants were aged between 13 and 17 years and assessed by an eating disorder specialist to ascertain that they met the Diagnostic and Statistical Manual of Mental Disorders (fifth edition (American Psychiatric Association, 2013)) diagnostic criteria for anorexia nervosa. Exclusion criteria were acute psychotic disorders or concurrent substance use disorders.

All participants in this study had previously undergone at least one prior outpatient treatment for anorexia nervosa, mainly of a multidisciplinary nature, which had proven ineffective due to either weight loss or insufficient weight gain after at least 8 weeks of treatment. They were therefore treated with intensive CBT-E, which consists of 13 weeks of inpatient treatment followed by 7 weeks of day hospital. Outcome measures were evaluated at baseline, at the conclusion of the 20-week treatment period and again at the 20-week follow-up. Patients admitted before the COVID-19 emergency were treated from 1 January 2016 to 31 April 2019, those during the COVID-19 emergency from 9 March 2020 to 31 March 2022 (the end of the COVID-19 emergency as declared by the Italian Government [13]) and those after the COVID-19 emergency from 1 April 2022 to 31 March 2023. All participants, along with their parent(s) and/or legal guardian(s), provided informed w 1ritten consent for their clinical data to be collected and anonymized within a service-level research setting. Ethical approval for the study was obtained from the GHC Institutional Review Board (Protocol Code 0013GHCIRB).

### 2.2. Treatment

Intensive CBT-E has been thoroughly detailed in prior publications [14,15,16]. In brief, however, admission is voluntary, with patients attending 3–4 preparatory sessions before admission. During the inpatient phase, patients participate in individual and group sessions led by a psychologist, focusing on CBT-E strategies and procedures, including those promoting weight gain. A CBT-E-trained dietitian assists with eating until the patient is able to finish meals without assistance. CBT-E-trained physicians and nurses oversee the patients’ physical health throughout the program. As the treatment progresses into the day-hospital phase, the focus shifts to preventing and managing potential setbacks. Accredited teachers provide lessons to ensure the adolescent patients’ schooling is not interrupted. Towards the end of the treatment, parents attend individual and joint sessions with the therapist and patient, being recruited as “helpers” in the creation of a positive, stress-free home environment in preparation for the patient’s return.

The treatment is traditionally administered in an “open” unit, with patients free to come and go under expert supervision, learning how to handle environmental stimuli that may exacerbate their eating disorder psychopathology. The unit is open, with the aim of reducing the high relapse rate commonly reported after discharge from closed units [17]. However, although no change in recruitment occurred, several restrictions were imposed when the national lockdown was initiated on 9 March 2020. Specifically, patients could no longer leave the unit or access the unit garden during the inpatient phase, and family visits were no longer allowed. Furthermore, in the day-hospital phase, individual and group sessions had to be delivered remotely, as did even core elements of CBT-E like collaborative weighing and assisted eating.

After the 20-week program, completers were offered 20 post-inpatient outpatient CBT-E sessions during follow-up, which is routine practice in the unit. In the post-inpatient outpatient step, a unified team of healthcare workers, all officially trained in CBT-E, focus on fostering relapse-prevention skills and addressing residual eating disorder features using CBT-E strategies and procedures. While the pre-COVID-19 group attended these sessions in-person, the during-COVID-19 group received them online. When the end of the COVID-19 emergency was declared on 1 April 2020, the intensive CBT-E program returned to being delivered in an open unit, reinstating all the pre-pandemic procedures.

### 2.3. Assessment

The study employed well-established and widely used outcome measures. Assessments were carried out at admission (baseline), at the conclusion of the 20-week treatment period (EOT) and during a follow-up session after 20 weeks. These measures included:

Body Weight and BMI-for-age percentile, the latter being calculated from the patient’s height and weight using the Center for Disease Control and Prevention growth charts [18] (http://www.cdc.gov/growthcharts/percentile_data_files.htm, accessed on 1 March 2024). BMI-for-age percentile with a value <1 was calculated as 0.5.

Eating Disorder Psychopathology, as assessed using the Eating Disorder Examination Questionnaire (EDE-Q, 6th edition, Italian version) [19]. The internal consistency in our sample was 0.95.

General Psychopathology, scored on the Brief Symptom Inventory (BSI), Italian version [20]. The internal consistency in our sample was 0.94. 

Clinical Impairment, assessed using the Italian version of the Clinical Impairment Assessment (CIA) [21]. The internal consistency in our sample was 0.96.

### 2.4. Outcome Categories

Two operational outcome categories were adopted and calculated for completers:“Good BMI outcome”, defined as the patient having achieved the lowest threshold in a healthy BMI range [22], namely the BMI-for-age percentile corresponding to an adult BMI of ≥18.5 kg/m^2^ [23];“Full response”, defined as the patient achieving both a BMI-for-age percentile corresponding to an adult BMI of ≥18.5 kg/m^2^ and a global EDE-Q score of less than 1 SD above the community mean (i.e., <2.77) [24], reported as a simple and replicable way of defining an excellent outcome [25].

### 2.5. Statistical Analysis

Descriptive statistics were applied as follows: means and standard deviations (SD) for continuous variables and percentages for categorical variables. Baseline differences among the three groups were assessed using analysis of variance with Bonferroni correction or the chi-squared test, as appropriate. Mixed-effects modeling was applied to analyze the effects of CBT-E on various outcome measures (BMI-for-age percentile, global EDE-Q, CIA and BSI scores) at both the end of treatment and at the 20-week follow-up across the different phases of the COVID-19 pandemic. The model incorporated fixed components, including the three groups, a time variable and the interaction between groups and time to evaluate changes over time. The random component of the model accounted for individual patient variations. Model fit was evaluated using three covariance structures of error terms, and a two-level model was constructed with time nested within individuals [26,27]. These models were designed to accommodate missing data patterns through the assumption of “missing-at-random” (MAR), supported by logistic regression analysis. Missing data were handled using multiple imputation procedures, generating five imputed datasets for pooled results [28,29]. Statistical analyses were conducted using SPSS software (IBM SPSS Statistics, version 28.0).

## 3. Results

### 3.1. Sample

A total of 132 adolescent patients with anorexia nervosa were included. Among those, 64 patients (48.5%) were recruited before, 37 (28%) during and 31 (23.5%) after the COVID-19 pandemic. The great majority of patients were female, and the mean age of groups was roughly 16 years. A comparison of the three groups indicated that they had similar demographic characteristics. Moreover, they had similar mean BMI-for-age percentile, eating disorder psychopathology and behavior, general psychopathology and clinical impairment scores. The demographic and clinical features of the three groups are presented in Table 1.

### 3.2. Follow-Up Completion

Fifty-four study participants completed the intensive CBT-E program before (84.4%), 27 (73%) during and 27 (87.1%) after the COVID-19 pandemic (chi-squared = 2.81, *df* = 2, *p* = 0.246). There were no significantly different baseline characteristics between completers and non-completers in any group. Among completers, 41 (41/54; 75.9%) of the pre-pandemic group, 17 of the during (17/27; 63%) and 21 of the post-pandemic group (21/27; 77.8%) attended the follow-up interview/assessment session at 20 weeks (chi-squared = 1.93, *df* = 2, *p* = 0.380). About 90% of participants (before n = 38/41, 92.7%; during n = 13/17, 76.5%; after n = 20/21, 95.2%) had received some form of treatment in the interim (chi-squared = 4.37, *df* = 2, *p* = 0.112). In 86.8% of before patients, 92.3% of during patients and 60% of after patients, this was a 20-week post-intensive CBT-E-based treatment delivered by trained therapists living close to their place of residence (chi-squared= 7.42, *df* = 2, *p* = 0.025).

### 3.3. Response to Treatment

Table 2 shows the BMI-for-age percentile, eating disorder and general psychopathology and clinical impairment at each time point, using intention-to-treat analysis with multiple imputations. The model fit was assessed by testing three distinct covariance-of-error term structures (unstructured, compound symmetry and first-order autoregressive) and evaluating three fit statistics (-2LL, AIC, BIC). Despite negligible differences among the fit statistics for the various structures, the unstructured covariance structure was chosen due to its superior fit, as is often observed. This structure, commonly encountered in longitudinal data analysis, is preferred for its parsimony and lack of assumptions in the error structure [30]. The data show that patients treated before, during and after the COVID-19 emergency had a similar increase in mean BMI-for-age percentile, which reached the normal weight range, and a clinically relevant improvement in eating disorder and general psychopathology and clinical impairment, both at the end of treatment and at the 20-week follow-up.

In examining trajectory patterns, all growth patterns consistent with quadratic shapes were found to be significant, but those consistent with cubic shapes were not. This suggests significant rates of change in both linear and higher-order trajectories for BMI-for-age percentile, EDE-Q global and subscale scores and BSI and CIA global scores.

The mixed model analysis indicated a significant change over time in all measured variables (Figure 1). However, no significant time–group interaction was found in the variables investigated (all ps > 0.05). Moreover, the three groups showed similar “good BMI outcome” and “full response” rates at both EOT and follow-up (Table 3).

## 4. Discussion

This study aimed to evaluate and compare the impact of an intensive CBT-E program on adolescents with anorexia nervosa across three distinct periods—before, during and after the COVID-19 pandemic emergency—and to evaluate post-pandemic treatment outcomes, yielded several findings. First and foremost, there were no discernible differences in eating disorder or general psychopathology or clinical impairment scores among adolescents across the three treatment periods. Likewise, dropout rates remained consistent across the groups, and all three cohorts exhibited significant improvements in BMI-for-age percentile from baseline to EOT, which were sustained up to the 20-week follow-up assessment.

Across all periods, nearly all adolescents achieved favorable BMI-for-age percentile outcomes at both EOT and the 20-week follow-up, with approximately 60% maintaining a full response at the latter time point. Notably, demographic characteristics and baseline psychopathology among groups were comparable, regardless of the phase of the COVID-19 emergency, suggesting that the severity of psychopathological features did not influence treatment outcomes during the pandemic, as compared to other periods. Likewise, these findings appear to suggest that the changes introduced during the lockdown phase (remote delivery, etc.) had no significant influence on the delivery of intensive CBT-E.

These findings are in line with previous studies comparing treatment outcomes in anorexia nervosa before and during the COVID-19 pandemic [11,12], but deviate from others [9,10]. These differences could be attributed to different treatment settings and different groups of patients (adults vs. adolescents). Some prior research, including on patients treated via intensive CBT-E, has suggested a more pronounced improvement among patients treated before the COVID-19 pandemic [8]. However, this disparity may be attributable to differences in the sample assessed before the pandemic. Indeed, although matched for age, gender, and BMI to the group treated during the pandemic, the pre-pandemic sample in the prior study included both adults and adolescents up to 16 years of age.

It is important to acknowledge the limitations of the present study and inherent in the small sample size of the three groups. Although a common problem of treatment outcome studies on anorexia nervosa, these may have precluded the detection of nuanced differences in treatment outcomes. Additionally, the results cannot be generalized to adolescent patients with anorexia nervosa treated in different and less intensive settings. However, our results provide further evidence of the efficacy of intensive CBT-E in treating adolescents with anorexia nervosa, even under challenging circumstances like the restrictions imposed during the COVID-19 pandemic. Furthermore, they suggest that patients treated after the pandemic did not experience prolonged effects in terms of treatment outcomes. With a commendable treatment completion rate exceeding 80%, accompanied by notable improvements in clinical indicators, the findings underscore the resilience and adaptability of this therapeutic approach. Notably, the sustained positive changes observed post-treatment persisted through the 20-week follow-up period, with nearly two-thirds of completers achieving a full response. These outcomes highlight the enduring impact of intensive CBT-E in promoting recovery and encourage the use of this treatment for adolescents with anorexia nervosa.

## 5. Conclusions

In conclusion, this study underscores the effectiveness of intensive CBT-E as a viable treatment option to offer to adolescents with anorexia nervosa, even during and after unprecedented challenges such as those posed by the COVID-19 pandemic.

## Figures and Tables

**Figure 1 nutrients-16-01411-f001:**
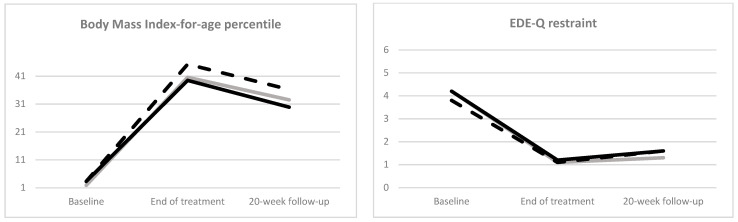
Estimated means of BMI-for-age percentile, subscales and global Eating Disorder Examination Questionnaire (EDE-Q), Brief Symptom Checklist (BSI) and Clinical Impairment Assessment (CIA) global scores at each time point in adolescent patients with anorexia nervosa treated before (grey continuous line), during (black dotted line) and after (black continuous line) the COVID-19 pandemic.

**Table 1 nutrients-16-01411-t001:** Baseline characteristics of adolescent patients with anorexia nervosa consecutively admitted to the unit either before (from 1 January 2016 to 31 April 2019), during (from 9 March 2020 to 31 March 2022) or after (from 1 April 2022 to 31 March 2023) the COVID-19 emergency. Data are presented as mean (SD) and [range] or as n (%).

Groups	Pre-Pandemic (n = 64)	During Pandemic (n = 37)	Post-Pandemic (n = 31)	*p*-Value
Gender, female, n (%)	63 (98.4%)	36 (97.3%)	31 (100.0%)	0.661
Other demographic and clinical characteristics, mean (SD) [range]				
Age, years	16.3 (1.4)	15.9 (1.6)	16.0 (1.3)	0.425
Body mass index-for-age percentile	1.9 (4.0)	3.3 (4.9)	3.3 (5.1)	0.226
Age of onset, years	14.3 (1.4)	14.1 (1.9)	13.7 (2.2)	0.330
Duration of illness, years	2.1 (1.0)	1.9 (1.2)	2.3 (1.9)	0.354
Eating Disorder Examination Questionnaire, mean (SD)				
- Global score	4.0 (1.4)	3.9 (1.4)	4.3 (0.9)	0.464
- Dietary restraint	4.2 (1.7)	3.8 (1.7)	4.3 (1.1)	0.372
- Eating concern	3.2 (1.4)	3.3 (1.5)	3.7 (1.0)	0.306
- Weight concern	4.0 (1.6)	4.1 (1.5)	4.4 (1.3)	0.549
- Shape concern	4.7 (1.4)	4.5 (1.6)	5.0 (1.1)	0.391
Eating Disorder Examination Questionnaire, n (%) if present *				
- Objective binge-eating episodes	22 (34.4%)	15 (40.5%)	8 (25.8%)	0.442
- Self-induced vomiting	11 (17.2%)	8 (21.7%)	7 (22.6%)	0.777
- Laxative misuse	6 (9.4%)	5 (13.5%)	0 (0.0%)	0.122
- Excessive exercise	42 (65.6%)	27 (73.0%)	24 (77.4%)	0.460
Brief Symptom Inventory, Global Severity Index	2.1 (0.9)	1.7 (0.9)	2.2 (0.6)	0.046
Clinical Impairment Assessment, Global score	34.0 (11.6)	30.9 (11.6)	37.3 (7.0)	0.058

* Number and percentage of patients who presented eating disorder behavior.

**Table 2 nutrients-16-01411-t002:** Mean and standard error (SE) of baseline, end-of-treatment and 20-week follow-up data in 132 adolescent patients with anorexia nervosa treated before, during and after the COVID-19 pandemic. Intent-to-treat analysis with multiple imputation procedure was used. Pooled data are presented.

		Mean (SE)		
	Sample	Baseline	End of Treatment	20-Week Follow-Up
Body mass index-for-age percentile	Pre-pandemic	1.9 (0.5)	40.6 (3.0)	32.5 (6.3)
	During pandemic	3.3 (0.8)	45.3 (5.2)	36.2 (9.5)
	Post-pandemic	3.3 (0.9)	39.5 (3.8)	29.9 (7.7)
Eating Disorder Examination Questionnaire				
Restraint	Pre-pandemic	4.2 (0.2)	1.1 (0.2)	1.3 (0.3)
	During pandemic	3.8 (0.3)	1.1 (0.2)	1.6 (0.4)
	Post-pandemic	4.2 (0.2)	1.2 (0.2)	1.6 (0.5)
Eating concern	Pre-pandemic	3.2 (0.2)	1.3 (0.2)	1.2 (0.2)
	During pandemic	3.3 (0.2)	1.4 (0.2)	1.6 (0.4)
	Post-pandemic	3.7 (0.2)	1.4 (0.2)	1.3 (0.4)
Weight concern	Pre-pandemic	4.0 (0.2)	2.1 (0.2)	1.6 (0.2)
	During pandemic	4.0 (0.3)	2.0 (0.3)	1.7 (0.4)
	Post-pandemic	4.4 (0.2)	2.1 (0.3)	1.7 (0.4)
Shape concern	Pre-pandemic	4.7 (0.2)	3.5 (0.2)	2.5 (0.3)
	During pandemic	4.5 (0.3)	3.1 (0.3)	2.5 (0.5)
	Post-pandemic	5.0 (0.2)	3.2 (0.3)	2.7 (0.5)
Global score	Pre-pandemic	4.0 (0.2)	2.0 (0.2)	1.7 (0.2)
	During pandemic	3.9 (0.2)	1.9 (0.2)	1.9 (0.4)
	Post-pandemic	4.3 (0.1)	2.0 (0.2)	1.8 (0.4)
Brief Symptom Inventory				
Global score	Pre-pandemic	2.1 (0.1)	1.1 (0.1)	1.0 (0.2)
	During pandemic	1.7 (0.1)	1.1 (0.2)	1.1 (0.2)
	Post-pandemic	2.2 (0.1)	1.3 (0.2)	1.4 (0.3)
Clinical Impairment Assessment				
Global score	Pre-pandemic	34.0 (1.4)	16.2 (1.7)	12.9 (2.1)
	During pandemic	31.0 (2.0)	15.6 (2.2)	15.7 (3.6)
	Post-pandemic	37.3 (1.2)	17.0 (2.3)	16.8 (4.4)

**Table 3 nutrients-16-01411-t003:** “Good BMI outcome” and “full response” at end of treatment and 20-week follow-up among adolescent patients with anorexia nervosa completing treatment before, during and after the COVID-19 pandemic. Data are shown as frequencies and percentages.

	End of Treatment				20-Week Follow-Up			
	Pre-Pandemic (n = 54)	During Pandemic (n = 27)	Post-Pandemic (n = 27)	Chi-Squared; *p*-Value	Pre-Pandemic (n = 41)	During (n = 17)	Post-Pandemic (n = 21)	Chi-Squared; *p*-Value
Good BMI outcome ^a^	54 (100%)	26 (96.3%)	27 (100%)	3.03; 0220	32 (78.0%)	15 (88.2%)	14 (66.7%)	2.52; 0.284
Full response ^b^	41 (75.9%)	18 (66.7%)	22 (81.5%)	1.63; 0.443	26 (63.4%)	11 (64.7%)	11 (52.4%)	0.85; 0.654

Note: ^a^ BMI-for-age percentile corresponding to an adult BMI of ≥18.5 kg/m^2^; ^b^ BMI-for-age percentile corresponding to an adult BMI of ≥18.5 kg/m^2^ and Eating Disorder Examination Questionnaire global score of less than one standard deviation above the community mean (i.e., below 2.77).

## Data Availability

The data presented in this study are available on request from the corresponding author due to ethical restrictions.

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
