# Peer review of "Intensive Cognitive Behavioral Therapy for Adolescents with Anorexia Nervosa Outcomes before, during and after the COVID-19 Crisis"

_nutrients, 2024, doi:10.3390/nu16101411_

Round 1

Reviewer 1 Report

Comments and Suggestions for Authors

Dear authors,

Thank you very much for your interesting manuscript. It's good to see the extent to which the success of your therapy programme before, during and after the Covid 19 pandemic has obviously been traceable almost independently of external factors. 

Please don't misunderstand me, but the results are presented very stringently and objectively; also, in my opinion, table 2 seems only of limited help in comprehensively understanding the relevance of your important results and appear very "academic". Would it be possible to present the results with a little more patient-related relevance?
The readership should be given a little more encouragement to use this therapy option for their patients in case of doubt. 

Thank you very much and 
Kind regards

Author Response

We thank the reviewer for their positive comments and the time dedicated to our manuscript. 

As suggested, in the Discussion section, we have underlined that the outcomes highlight the enduring impact of intensive CBT-E in promoting recovery and encourage the use of this treatment for adolescents with anorexia nervosa.

Reviewer 2 Report

Comments and Suggestions for Authors

This is a well written paper focused on: Intensive cognitive behavioral therapy for adolescents with anorexia nervosa outcomes before, during and after the COVID-19 crisis. 

I do have one revision/edit:  on lines 96-97 the dates included do not seem to be correct? 

Author Response

We thank the reviewer for the positive comments

As suggested, we have corrected the typos on the (lines 96-97) dates.

Reviewer 3 Report

Comments and Suggestions for Authors

Dear Author 

thank you for sharing this research.

This study assesses the impact of intensive CBT-E on adolescents with anorexia nervosa before, during, and after the COVID-19 pandemic. Findings reveal consistent improvements in eating-disorder and psychopathology scores across all periods, with sustained benefits up to a 20-week follow-up. Demographics and baseline psychopathology remained comparable regardless of pandemic phase, suggesting treatment efficacy was unaffected. Although limited by sample size, results support intensive CBT-E's effectiveness even amid pandemic challenges. Treatment completion rates exceeded 80%, with enduring positive changes observed post-treatment. These findings underscore the resilience of CBT-E in promoting recovery for adolescents with anorexia nervosa, offering lasting benefits.

I found it well-written and I have no change to request. 

Best Regards

Author Response

We thank the reviewer for the time dedicated to our manuscript and for their positive comments.